# “Wounds Home Alone”—Why and How Venous Leg Ulcer Patients Self-Treat Their Ulcer: A Qualitative Content Study

**DOI:** 10.3390/ijerph16040559

**Published:** 2019-02-15

**Authors:** Mirna Žulec, Danica Rotar-Pavlič, Zrinka Puharić, Ana Žulec

**Affiliations:** 1Medical Faculty, Department of Family Medicine, University of Ljubljana, Poljanski nasip 58, Ljubljana 1000, Slovenia; danica.rotar@gmail.com; 2Study of Nursing, Bjelovar University of Applied Sciences, Trg. E.Kvaternika 4, Bjelovar 43000, Croatia; zpuharic@vub.hr; 3Polyclinic Marija, Kneza Mislava 2, Zagreb 10000, Croatia; ana.zulec@gmail.com

**Keywords:** chronic wound, venous leg ulcer, self-treatments, nurse, home care nurse, informal caregiver

## Abstract

Background: Venous leg ulcers (VLUs), the most common type of leg ulcerations, have long healing times and high recurrence rates; reimbursement rules and a general shortage of nursing staff have put self-treatment into focus. The study aimed to investigate why and how patients with VLUs self-treat their ulcers. Methods: Patients with VLUs (*N* = 32) were selected by criterion sampling for a multicentric qualitative study using semi-structured interviews. The interviews were analyzed via inductive qualitative content analysis. Results: More than two-thirds of participants sometimes self-treated VLU and one quarter changed their prescribed treatment. Experiences were expressed through four themes as follows: (a) current local VLU therapy; (b) VLU self-treatment; (c) patient education; and (d) psychosocial issues. The main reasons for self-treatment were a lack of healthcare resources, reimbursement restrictions, and dissatisfaction with conventional treatment together with insufficient knowledge about the wound-healing process and possible side effects. No educational materials were provided for patients or caregivers. Many patients adopted homemade remedies. Conclusion: Patients with VLUs practice self-care due to limited healthcare availability, a low awareness of the causes of their condition, and the effects of therapy on VLU healing. Future educational intervention is needed to enhance self-treatment.

## 1. Introduction

A venous leg ulcer (VLU) is an open skin lesion that usually occurs on the medial side of the lower leg between the ankle and the knee as a result of chronic venous insufficiency (CVI) and ambulatory venous hypertension, and that shows little progress towards healing within 4–6 weeks of initial occurrence [1]. VLUs constitute the most common type of leg ulceration, affecting approximately 1% of the whole population, and 3% of those >80 years old, in Westernized countries [2]. In addition to their long healing time, the 12-month recurrence rate of VLUs ranges from 18% to 28% [3]. VLUs have a significant impact on quality of life, entailing personal, social, and psychological effects, with broad social and economic impacts [2]. Treatments for these ulcerations are well documented and founded on accurate diagnosis, appropriate local ulcer care, and the application of sustained, graduated compression therapy [2,3,4,5,6]. VLUs can be treated by general practitioners, home-care or community nurses, hospital wards, and specialized clinics.

The self-treatment of wounds is part of a self-care concept that has many definitions, but mainly focuses on the ability of individuals, families, and communities to promote and maintain health, prevent disease, and cope with illness and disability, with or without the support of healthcare providers [7,8,9]. Little research has been done on the self-care of chronic wounds, particularly VLUs, with the exception of a few articles on exercise and lifestyle changes after recovery from VLU [10,11]. Some studies have investigated self-care with Orem’s theory as a framework [12,13], compression and skin care in patients with VLUs [14], or overall self-management in self-care [15] have been a focus. Recent studies have found that patient independence is the main factor in self-treatment [10], as is distrust of the healthcare professionals that provide advice and care ([11]. Standardized education and supervision are recommended to optimize self-care practice [16]. As in other chronic diseases, patient compliance is necessary to achieve VLU healing and prevent recurrence. Several researchers have found that improved compliance leads to enhanced diagnostic effectiveness and disease treatment, with considerable economic benefits [17,18]. These days, the patient—health care provider relationship has a wide scope, ranging from compliance [19,20,21] to concordance, as a shared decision-making process [22,23]. However, as a consequence of non-compliance, lack of access to healthcare resources, or other objective or subjective factors, patients often self-treat their wounds [24,25].

To ensure consistency in VLU management and continuity of care, patients must understand their illness, have knowledge of basic wound-care principles, and practice other self-care activities related to VLUs as a chronic disease. Quality health knowledge contributes to increased awareness of health care, which improves patients’ self-care abilities and helps them avoid unhealthy behaviors and adopt healthy lifestyles. Specific patient characteristics, such as educational level, occupation, rural/urban residence, and region of residence, are associated with knowledge of chronic diseases and the development of self-management behaviors [26,27,28,29,30,31].

In Croatia, patients must first visit their general practitioner and then be referred to a specialist. The general practitioner decides whether a home-care nurse can perform dressing changes (if the patient has a mobile impairment or lives far from the medical facility) or the patient should go to a general practitioners’ office for this procedure. Home-care nurses gain vocational training during secondary school and are supervised on a monthly basis by a community nurse with a bachelor’s degree. Reimbursement rules state that only specialists can prescribe compression therapy and evaluate the use of silver dressings and stop or prolong its use. Because of this, most of VLU patients visit specialists every 1–3 months. The costs of wound dressings are reimbursed for 10 pieces per wound per month; if there is a change in wound status, a patient cannot receive a new dressing for at least 1 month. Negative pressure wound therapy is not reimbursed and is not used in a home setting.

As only a few studies have investigated the experiences of VLU patients with self-treatment, this study focuses on how and why patients practice self-treatment.

Self-treaters in this study were considered as patients who had VLUs and changed their wound dressings by themselves, on a regular or irregular basis, whether they were following the prescribed treatment or were changing it.

## 2. Materials and Methods

### 2.1. Design

A qualitative content study was conducted to determine why and how VLU patients practice self-treatment. The study was proceeded at the vascular surgery outpatient clinics of three hospitals in Croatia: The University Hospital Dubrava in Zagreb and two general hospitals in central Croatia in the towns of Koprivnica and Bjelovar. The researcher visited each outpatient clinic on randomly chosen days (1 day per week) during a 2-month period.

### 2.2. Sampling

Thirty-two participants were purposively selected by criterion sampling to include patients from both urban and rural areas [32]. Criterion sampling has the following characteristics:Can be useful for identifying and understanding cases that are information-rich;can provide an important qualitative component to quantitative data. There are lots of issues in VLU; andcan be useful for identifying cases from a standardized questionnaire that might be useful for follow-up [33,34].

We chose to conduct research in the clinics because almost all patients visit a clinical specialist on average every 3 months, mostly because reimbursement rules define that a clinical specialist must approve the use of a silver dressing. Visiting family doctors’ offices would require a considerable amount of time and an increasing number of researchers. Additionally, each of the clinics should give ethical approval for research, as most of these doctors are private, and in Croatia, no single body approves individual research, but gives it to every institution for itself. Also, outpatient clinics were chosen for data collection as patients referred from different general specialists and from different geographical areas could be interviewed.

In each clinic, there are a few days in a week for chronic wound patients or specialists who are in charge of them. The researcher randomly picked days for research during the research period.

Patients with VLUs were invited to participate in the study by a member of the usual healthcare team, who introduced the researcher to the patient. The researcher estimated the criteria for eligibility, introduced the patient to the informed consent, and offered the patient time to ask questions. After the patient signed the consent, the interview began. Interviews were performed in a separate private room and lasted 26–51 min.

Participants were eligible for inclusion if they had been diagnosed with one or more VLUs at least 3 months before the beginning of the study, were older than 18 years old, and were able to give informed consent. Patients who could not communicate reliably, who had a cognitive impairment, physical inability, or had a history of mental illness were excluded.

### 2.3. Data Collection

A literature review was undertaken to develop the interview framework, which was used as a guide for the semi-structured interviews (MŽ, DRP). It was redesigned after a pilot interview (MŽ, DRP). In-depth semi-structured interviews were conducted with participants between December 2016 and November 2017. MŽ conducted the interviews, which were performed until information saturation was achieved for each hospital. The University Hospital Dubrava provided nine patients, the hospital in Koprivnica had 12 patients, and there were 11 patients from the hospital in Bjelovar. To acquire insight into the patients’ therapeutic condition, each interview began with questions on demographic data following open-ended questions asking patients to characterize their current wound-therapy regimen, give the reasons they had adopted it, explain their ulcer self-treatment procedures, information sources, and describe the role of informal caregivers. The participants were encouraged to describe their experiences in self-treatment, to reflect on their involvement in the process, and to identify any facilitating factors or barriers. Different follow-up questions were asked depending on the answers to these questions.

### 2.4. Ethical Considerations

The overall ethical principles for this study were drawn from the World Medical Association’s Helsinki Declaration (2008). Ethical approval was obtained from the ethics committee of each participating hospital. All participants were given a verbal and written description of the aims of the study and their role. Written informed consent was obtained from each participant, and they were given assurance of the confidentiality and anonymity of their responses. They were also reassured that their care would not be affected in any way by their decision to participate in an interview, and all participants were assured that they would be able to withdraw from the study at any time.

### 2.5. Data Analysis

The interviews were recorded and transcribed verbatim. Data from the interview transcripts were coded, and then a comparative analysis of the codes was undertaken by the researchers independently (MŽ, ZP). Next, the codes were organized into subthemes and themes. Where consensus was not immediately attained, coders discussed the differently perceived parts until agreement was achieved [35]. Most of the authors have extensive experience in wound care, MŽ as wound care nurse, DRP as a general practitioner, ZP as a public health specialist, and AŽ has a degree in psychology and supports wound patients in a private outpatient wound clinic. MŽ and DRP have extensive experience using qualitative research methods. None of the researchers are employees of the hospitals where research was conducted.

To obtain rigor, the Consolidated Criteria for Reporting Qualitative Research guidelines [36] and the recommendations for the design of case study research in health care using the DESCARTE model [37] were followed, as well as criteria for establishing the trustworthiness of the data by reviewing issues concerning data credibility, transferability, dependability, and confirmability [38].

## 3. Results

In total, 32 participants were interviewed, their demographic characteristics and wound durations are shown in Table 1.

After the transcript analysis, four categories and nine subcategories were identified (Table 2).

### 3.1. Theme 1: Current Local VLU Therapy

Patients were asked to explain their experiences with current local treatment. Within this theme, the following subthemes were organized: The conduct of current local VLU therapy, compression therapy, and effective therapy. 

#### 3.1.1. Subtheme: Conduct of Current Local VLU Therapy

When asked to describe their current local wound therapy, the patients reported ulcer irrigation and dressing application. The researchers identified 29 codes. Codes are presented in Appendix A. The patients described dressing changes done by home-care nurses. When asked about peri-ulcer skin care, patients mentioned several different creams.
“The nurse came every other day and she changed it. Every other day she came, made it clean and tidy and everything was covered.” (Identification number (ID)4, female (F), wound duration (WD) 6 years)
“The nurse comes to me.... she rinses it with a solution and puts the dressing on.” (ID28, male (M), WD 1.5 years)

#### 3.1.2. Subtheme: Compression Therapy

The participants referred to compression therapy as an additional, rather than a vital, part of therapy. The majority used long-stretch bandages, few wore compression stockings, and very few applied multi-layer compression. They usually performed compression therapy themselves. They also expressed that they had trouble wearing it during everyday activity. Details regarding compression therapy type are shown in Table 3.
“Well, it has to do with the sock. It rolls down and you just can’t... And sometimes it pinches a lot. It gets tight, so I have to unwind it because it gets really tight. Once I unwound it in the park... I took the wound dressing off in the park because it had tightened, I couldn’t walk any longer... underneath the sock, when I take it off, it starts itching like hell. And my whole leg starts to itch. Because it’s tight the whole day.” (ID18, F, WD 3 years)
“Well, I wear bandages all the time. When I wear my boots, it gets sweaty in the heat.” (ID16, F, WD 7 years)
“In the beginning, maybe only a couple of times, but ever since, it’s so normal and it’s easy for me to walk in it.” (ID19, F, WD 3 years)

#### 3.1.3. Subtheme: Effective Therapy

The patients demonstrated an inability to determine what was an effective treatment for their VLU. Lack of confidence in the treatment and despondency were reflected by their responses.
“To be honest, I don’t know what will help me. Today they start with this; they start with that, and a week later it is another thing.” (ID19, F, WD 3 years)

Some of them came to their check-ups just because they had an appointment, not because they believe in effectiveness of therapy, so the waiting room is a sort of social event for them.
“I don’t know. I came here because I had an appointment.” (ID12, M, 15 years)

### 3.2. Theme 2: VLU Self-Treatment

Patients mainly took care of their wounds on the weekends, but also when the pain or itching was so disturbing that a dressing change was the only way to get relief. Dressing changes, as they saw it, should be performed by a home care nurse or doctor. Patients were confident in the way they changed their dressings, but had different approaches to wound irrigation and this reflected inconsistencies among the recommendations they had received from medical professionals. They were worried of whether showering the ulcer was allowed; some of them were surprised when the researcher asked this question. This theme included the following subthemes: Performing VLU self-treatment and changing a prescribed treatment. Codes for this theme are presented in Appendix A.

#### 3.2.1. Subtheme: Performing VLU Self-Treatment

About 70% of the participants self-treated their ulcer at some point, and half of them had occasionally self-treated their wound during the 30 days before the interview. They were asked to describe how they changed the dressing.
“First, I wash my hands so they’re clean. I have clean gauze and then I clean it [the ulcer]. I can do this professionally, as a nurse does! And then I put on the clean gauze, then a thin bandage, the elastic one, and that’s it!” (ID19, F, WD 3 years)

A few patients, mainly those in rural areas, did not have a nurse for home-care visits, so they practiced self-treatment the entire period that they had their ulcer.
“The doctor didn’t give me anyone else, no. He gave me an ointment, but he couldn’t get anyone else to do it. I did it, and my husband helped me when he could.” (ID4, F, WD 6 years)

Patients also had issues with wound dressing side effects.
“I can’t stand that [the wound dressing] in a shiny envelope… It’s pinching the whole leg…It was very good when it was with gauze, but the doctor told me to put that…And the wound wasn’t dry anymore, it was just horrible…and it’s not good to keep it covered all the time.” (ID18, F, WD 2 years)

Some patients said that ulcer care should be conducted exclusively by a doctor or nurse, and they expressed a fear of touching their ulcer.
“I’m afraid to take care of it or to touch it.” (ID27, M, WD 6 years)

#### 3.2.2. Subtheme: Changing the Prescribed Treatment

The participants were asked to explain whether and why they changed their treatment. They were surprised at this question at first, but then, about 20% of participants admitted that they had changed their treatment and tried some other options. The main reason for changing treatment was due to slow wound healing. These patients do not have access to medical information, other than the opinions provided by family, neighbors, and/or TV. Thus, they did not use an alternative modern dressing or seek help in changing the healthcare provider. When asked why they did not change their doctor, they verbalized a fear of being rejected by the doctor they left if they were to seek help in the future from that doctor, as well as a lack of information on where else to go.
“I don’t know where to go? Where?” (ID19, F, WD 3 years)

As replacement therapy, they used what they heard from mentioned resources, mainly folk remedies. Thus, several codes that included homemade remedies were identified: Vinegar cream, homemade pork fat and marigold cream, sour weed (*Rumex acetosella*) cream, hazel dormouse (*Muscardinus avellanarius*) fat, and smashed eggshell. The mixtures were applied topically to the VLU or peri-ulcer skin.
“I took that vine cream, which cleaned it.” (ID13, F, WD 10 months)
“I take 40 marigold flowers. I don’t wash them. I just wipe them very well and put in four spoonfuls of pork fat, but pure fat. It mustn’t be buttery. Then I warm up the fat with the flowers in it. And I mix it over a small fire, so the fat becomes yellow. And then I decant it through the gauze and I have the fat. I put it in the refrigerator. I apply it around the cracked part of the wound. I read about it somewhere.” (ID21, F, WD 2 years)
“Then I put on the badger fat and the skunk fat.” (ID26, F, WD 10 months)
“Sometimes I apply some honey on the wound and around it, yeah, that too. And then I heard about the grass as well. The one with sorrel, as they say. That wide leaf. I put it on myself and now I have a very large wound.” (ID24, F, WD 30 years)

Sometimes, they changed prescribed treatments, for example, from wound dressings to saline solution dressings, due to discomforts, such as sensations of burning or tension.
“I preheated some water. I put in some salt, that’s what the doctor told me, and I put it on the ulcer. The gauze too, they told me.” (ID24, F, WD 30 years)

### 3.3. Theme 3: Patient Education

Patients were asked to explain where they got information about VLU management and how they followed instructions if received. In this theme, two subthemes were identified: Receiving information and following instructions.

#### 3.3.1. Subtheme: Receiving Information

Patients were questioned about their source of information on local wound care compression therapy, signs of worsening of the wound, and how to avoid risk factors or unhealthy behaviors. In general, no patients received written instructions or any other patient material from clinics or general practitioners; they only received verbal instructions or simply learned by observing their home-care nurse.
“The doctor explained it to me and then I remembered it.” (ID20, F, WD 2 years)
“Well yes… sometimes one would read a thing or two [meaning educational leaflet]… you’re scared all the time… why is it yellow, why it hurts.” (ID5, F, WD 5 years)

#### 3.3.2. Subtheme: Following Instructions

Patients used ulcer-care procedures that they saw at the hospital or that were performed by a home-care nurse.
“I saw my nurse wrap it for me, and then I did it the same way.” (ID21, F, WD 2 years)
“At first, I wrapped it with the bandage on my own and then I didn’t know how to handle it. I stopped it. But then the doctor... told me about the sock. But nobody showed me how…how to put it on.” (ID27, M, WD 6 years)

They were asked if they thought that some kind of guide would help them. They all agreed that it would be beneficial, especially to instruct them what to do when their wound itched or when to seek help. They also expressed interest in peri-ulcer skin care and explanations regarding wound showering.

### 3.4. Theme 4: Psychosocial Issues

In this theme, patients were asked to explain how significant others involved in ulcer care were and if there were any issues (e.g., long wait times, bureaucracy, anything they experience) related to wound care. Two subthemes were identified: The role of informal caregivers and fear of the future.

#### 3.4.1. Subtheme: Role of Informal Caregivers

The family was rarely involved in dressing changes; they usually transported patients to the doctor or did some paperwork instead. When they were included in actual care, it was often women in the family or spouses. Patients thought that wound care is their problem and do not want to burden anyone with it. Table 4. describes overview of informal care givers involved in self-treatment.
“My husband helps me. He washes his hands and puts on gloves and then he irrigates the ulcer. And he unwinds it after a day or two, for instance, when my nurse isn’t working or something, and then he does it, as I direct.” (ID5, F, WD 5 years)
“My daughter lives next to me, she’s a nurse and she makes the dressing change.” (ID20, F, WD 2 years)

A few (all men) angrily stated that nobody was involved in the care of their wound.
“No, nobody takes care of my ulcer. Who should take care of it [angrily]? I don’t want that; it’s my ulcer!” (ID22, M, WD 26 years)

#### 3.4.2. Subtheme: Fear of the Future

Finally, the interviewer asked the participants about any problems related to wound care, including social aspects, health care system problems, and personal issues. However, the main concern was the fear of the future. Patients were mainly afraid about how their wound would ultimately affect them, and if it would ever heal, or if their limb would eventually be amputated. In the case of amputation, they worried over how they would care for their ill spouses; those who lived alone concerned over the prospect of having to move to a nursing home and were fearful about the difficulties of life after amputation.
“I’m frightened if this ulcer will be bigger…if not heal…I’m really scared of it…that I’ll not succeed…I’m really frightened.” (ID24, F, WD 5 years)
“What worries me? You can imagine what…you know what will be if this won’t heal [showing cutting the leg]… [crying] (ID4, F. WD 6 years)
“I was worried about when I got well and took a bath my way. (laughter) To lie in the bath…And to put on anything and everything… (laughter) And you know how I take baths now? Like a stork. I lift one leg in the air like this, and then that’s how I bathe.” (ID6, F, WD 2 years)

## 4. Discussion

This research qualitatively explored VLUs patients’ self-treatment practice to understand their reasons to do it, sources of information, and psychosocial issues that affect them. Participants in this study self-treated their VLUs for both subjective and objective reasons, and self-treatment was both short- and long-term.

Objective reasons included a lack of healthcare resources (e.g., no home-care nurse over the weekends or no home-care nurse at all) and reimbursement restrictions (e.g., insufficient dressings for heavily exuding ulcers). From a subjective perspective, some participants followed medical recommendations until they became dissatisfied with conventional treatment, as caused by two factors: Dissatisfaction with the outcome of medical treatment and/or dissatisfaction with the physician or another responsible healthcare provider, consistent with a previous study [39]. When they were dissatisfied, patients followed the prescribed treatments, but performed the dressing changes themselves; they also switched between medications or deviated from the prescribed treatment. Dissatisfaction with medical treatment was observed when the possible side effects of treatment were not made clear to patients, leading to them being frightened by sensations in the ulcer area, and prompting them to conclude that the best solution would be to maintain traditional VLU management, sometimes supplemented with folk remedies. In addition, pain or other distressing symptoms, such as peri-ulcer skin itch or excessive exudation that they could not support, led to a change from prescribed treatment. These findings are in opposition to other studies that have found that patients choose self-treatment as a step forward toward independence and comfort [24,25].

Patients usually used home remedies and medicines that are well established in the geographic area, mainly marigold and aloe vera, whose benefits are well studied [40,41,42,43,44,45,46,47,48] Concern about the potential side effects of pharmaceuticals and an affinity for holistic approaches are predictors of engaging in self-care [49,50,51]. Hence, when clinicians have doubts about appropriate wound dressing [52], it is not surprising that patients will seek other possibilities. Compression therapy tended not to be emphasized, which in combination with other obstacles resulted in a long healing time. Indeed, the average healing time of VLUs in Croatia is 37 months [53]. For ulcer healing and recurrence prevention, life-long wearing of compression hosiery is a must; however, concordance with this is a problem for many patients [54,55,56].

This study revealed poor patient knowledge regarding their condition. For example, none of the patients mentioned exercise as a positive factor in wound healing: Instead, they apologized that they did not rest because they had much work to do related to housekeeping, working on a family farm, or taking care of an ill partner. VLUs patients are encouraged to perform self-care activities, such as increasing their physical activity and mobility levels, performing foot exercises and elevating their limbs, the assumption being that healing outcomes may be improved or recurrence rates reduced. Exercises, such as heel raises, flexion, extension, and rotation of the ankles, have been shown to be beneficial as they increase venous return [57,58,59].

There is no standardized VLU treatment protocol on national, primary, secondary, and tertiary level, clinical medical doctors decide according to their education. Therefore, clinicians and members of their teams verbally explain VLU treatment procedure to patients, with written specialist findings only mainly being found in communications with a family practitioner. No patients received written instructions on skin care, the importance of compression therapy, or any other aspect of treatment, and no patients received any other form of VLU education. Hence, it is necessary to raise awareness of the educational needs of this population if adequate and effective educational interventions are to be developed. Self-care knowledge and skills are often required for successful management of specific health states; to perform a self-care action for a specific person, one must possess knowledge of the action and its relation to a desired health response [12,60].

Because they did not receive any written information, except prescriptions for wound dressings, and insufficient oral instructions, the patients tried to replicate the procedures performed in a hospital or by the home-care nurse. The patients were emotionally connected with their home-care nurses, with whom they shared everyday ups and downs; in this way, home-care nurses represent role models for self-treatment and sources of local wound-treatment procedures. Previous qualitative studies have also reported that the nurse—patient relationship is important and positive with reflections that this was one of the only positive aspects of VLU [61,62]. Indeed, those studies indicated that nurses often go beyond the necessities of their visits. In our study, we found similar results, as a nurse was sometimes the only person that a patient saw in a few days. In rural areas, nurses are a connection to the outside world, and they may do things beyond their job, such as shopping for necessities. As patients may not be able to assess the quality of a nurse’s dressing changes, they may decide to simply put their faith in their nurses and maintain good relationships in the hope that their work is effective, but also because they represent more than just a healthcare worker to them. This may be the reason why we did not find more dissatisfaction with the care provided by nurses that some other research groups have reported. However, neither the home-care nurses nor the community nurses receive formal education on wound care and there is no difference in wound-care knowledge between bachelor’s degree students with or without any previous medical education [63]. Indeed, our findings are similar to those of other studies on populations in other countries [64,65], and this issue (the lack of a clear, long-term home-based strategy for wound care in VLU patients) is due to inconsistent nursing education and specifically wound-care education in nursing. Inconsistency in nursing education, lack of both patient understanding and provider expertise can promote an increased risk for patient nonadherence due to unintentional acceptance of antiquated treatment theory and a potentially less than fully reasoned or effective approach to the wound treatment plan of care [66,67].

In Croatia, nurses are not allowed to prescribe treatment or wound dressing; they can only follow doctor’s recommendations. Also, doctors rarely proceed with a wound dressing change, it is done mainly by nurses or patients. However, previous research suggests that that the organization of care rather than the location of care is most important.

Although, at some point, many patients did not follow recommendations closely, those patients did not want to give up the safety of the health system, and they visited their specialists as scheduled. To improve wound-healing times and the quality of life of these patients, it is necessary to take advantage of the fact that patients still adhere to the healthcare system and to educate them, preferably through a carefully planned interdisciplinary approach involving all levels of the healthcare team.

The role of informal care in wound self-treatment remains insufficiently investigated. It is increasingly evident that family members play an important role in the provision of home healthcare [68]. However, few studies have focused on the experiences of informal caregivers of people living with chronic wounds [69,70], although it has been found that more than one-third of wound care is conducted by caregivers [71,72,73]. These caregivers typically have a low quality of life and are overburdened [74,75]. Managing wound care symptoms can be challenging [63] and may lead to numerous health risks, such as the development of sleep disorders, anxiety, and depression, and economic consequences [68]. For our patients, informal caregivers enjoyed the highest level of trust, as they were generally family members and sometimes had a medical background. Thus, knowledge of and information on wound care is necessary for caregivers, as it is necessary for informed decision making. Informal caregivers for wound care need more significant information and training on the treatment of the object of their care [76].

VLU patients in general experience a lower quality of life [77,78,79,80], but their main concern was primarily wound healing and fear of amputation and losing mobility. Those who are especially affected are female patients, older adults, and patients who live alone, the predominant characteristics of the participants in this study. However, most participants thought they were old and sick and were surrendered to the disease as such, especially if their family had someone with a venous ulcer. We found that older couples were more connected and took care of each other, while younger patients expressed more anger or loneliness, similar to a previous study [81,82]. This possible trend could be a theme for future investigations.

However, deep inside, their main concern was primarily wound healing, and fear of amputation and losing mobility, consistent with other studies [25,83].

This fear was sometimes intensified by living in a rural or remote environment, which presents unique challenges for people with chronic conditions, mainly those created by limited healthcare services and physical and emotional isolation. However, although distance, isolation, weather, and transportation present obvious challenges, some qualitative research has also found that such factors can also have a positive impact on a patient’s social environment, as people living in rural areas tend to form stronger connections with neighbors, which can help make chronic conditions more tolerable [84].

A few limitations to this study need to be pointed out. First, the fact that these results cannot be extrapolated to all patients with VLU. Apart from differences in the healthcare system, health insurance, and medical staff training, only a few participants self-treated their wounds throughout the entire study period as it was challenging to recruit patients who self-treat their wounds. Such patients rarely, if ever, go to the doctor, making it nearly impossible to identify and reach them. While one previous study on this topic attempted to get around this challenge through Internet advertising [10], our patients generally do not use the Internet or computers. Hence, we decided to go to clinicians’ offices to identify patients who go against prescription and self-treat their wounds. Also, we went to three different hospitals to improve recruitment. By making the study multicentric and by randomly choosing interview dates, our results should not be affected by selection bias.

In addition, during interviews with some participants, patient admiration and satisfaction with doctors were apparent, and those patients were concordant with therapy. Those patients were collaborative and followed instructions, but the effects of compliance and concordance were not primarily explored in the research.

Further research should consider country-specific factors, such as climate, family culture (as it relates to chronic leg ulceration and across age groups), reimbursement rules, and especially nursing education. Finally, future work should include the development of an education program to help people self-treat chronic wounds with an instrument for evaluating one’s capacity for self-care. In this same vein, more research regarding the role of informal caregivers would also be beneficial to gain deeper insight in this area.

However, as formative research for future educational and intervention activities, it is essential to explore influential factors specifically in the context in which the intervention will be implemented [81].

## 5. Conclusions

This is the first study to describe VLU patients’ experience with self-treatment. Almost 70% of patients practiced self-treatment and a quarter of them changed prescribed treatment; this was due to limited healthcare availability, low awareness of the causes of their condition, and the effects of therapy on VLU healing. However, our sample was limited to only 32 patients from three different hospitals, and future research on this topic is necessary. Future initiatives to improve self-treatment among VLU patients should involve interventions, particularly educational programs.

## Figures and Tables

**Table 1 ijerph-16-00559-t001:** Overview of participants’ characteristics and wound duration.

Characteristics		
Age (*y ^1^*)		
Mean	68.2	
SD	7.6	
Range	59–81	
Sex (*n ^2^*, %)		
Male	12	37.5
Female	20	62.5
Wound duration (*n*, %)		
< 6 months	6	18.8
6 months–2 years	7	21.9
2–5 years	6	18.8
>5 years	13	40.6
Wound recurrence		
Yes	11	35
No	21	65

^1^ y = year, ^2^
*n* = number.

**Table 2 ijerph-16-00559-t002:** Overview of the subthemes and themes describing the experiences of VLU patients with self-treatment.

Subthemes	Themes
Conduct of current local VLU therapy Compression therapy Effective therapy	Current local VLU therapy
Performing VLU self-treatment Changing the prescribed treatment	VLU self-treatment
Receiving information Following instructions	Patient education
Role of informal caregivers Fear of the future	Psychosocial issues

**Table 3 ijerph-16-00559-t003:** Overview of participants use of compression therapy.

Answer	Frequency	Percent
Yes—elastic bandage	14	43.8
Yes—compressive sock	8	25.0
Yes—multilayer compressive	5	15.6
No	3	9.4
No, contraindicated	1	3.1
No answer	1	3.1

**Table 4 ijerph-16-00559-t004:** Overview of informal care givers involved in self-treatment.

Answer	Frequency	Percent
Lives alone	2	6.3
Daughter	4	12.5
Family member (not specified)	4	12.5
Sister-in law	1	3.1
Spouse	3	9.4
Did not want to involve and burden the family	9	28.1
None	7	21.9
No answer	2	6.3

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
