# Peer review of "“Wounds Home Alone”—Why and How Venous Leg Ulcer Patients Self-Treat Their Ulcer: A Qualitative Content Study"

_ijerph, 2019, doi:10.3390/ijerph16040559_

Reviewer 1 Report

As the authors mentioned in the abstract, the aim of this study is to investigate why and how the patients with venous leg ulcers (VLUs) treat their ulcers by themselves. The authors summarized the interview results of 32 patients with VLU including four themes: a) current local VLU therapy; b) VLU self-treatment; c) patient education; and d) psychosocial issues. The study topic is good, but the sample size (patient numbers) is small, and the following questions should be addressed.

1.      Is there any standard protocol of VLU treatment for the patients found in the hospital or somewhere? Or the clinical medical doctors have provided the details of the VLU treatment procedures to the patients?

2.      Is there any difference of the VLU healing results between the different treatments (the wound was treated by the doctors/nursing staff or patient self?

3.      Are dressing types the same used for all the patients or not? How many types of the dressing were used by the patients? Did the different dressings show the different healing results?

4.      How about the age effect on self-treated VLU? (The younger patients did better or worse?)

5.      Table 1 should be re-do. Please use “.” To instead of “,” in all the numbers in Table 1, such as “68,2” should be “68.2”; “7,6” should be “7.6”; “37,5” should be “37.5”; “62,5” should be “62.5”; ect.

6.      The authors should give their suggestions about self-treatment. I am just wondering if the authors recommend the VLU patients to self-treat the VLU? What is the authors’ suggestion?

7.      The study results should be described by more tables or graphs, rather than text description.

Author Response

Response to Reviewer 1 Comments

Point 1: As the authors mentioned in the abstract, the aim of this study is to investigate why and how the patients with venous leg ulcers (VLUs) treat their ulcers by themselves. The authors summarized the interview results of 32 patients with VLU including four themes: a) current local VLU therapy; b) VLU self-treatment; c) patient education; and d) psychosocial issues. The study topic is good, but the sample size (patient numbers) is small, and the following questions should be addressed. 

Response 1: We would like to thank the reviewer for all comments, and for giving us the chance to elaborate further on our project. The comments are highly insightful and have enabled us to improve the quality of the contents greatly.

We have conducted with definition of data saturation according to Guest [1].

 “Data collection and analysis should continue to a point when additional input from new   participants no longer changes the researchers’ understanding of the concept. This is the point of data saturation. In this study, we recruited interviewees strictly in line with the information saturation principleand we stopped interview when there was no more information about the research theme.

 According to Coyne (reference number 32 in our manuscript) “A purposive sample size was      

 determined on the basis of the theoretical saturation, which is the point in the data collection

 process when new data no longer bring offer additional insights to the research question.”

Point 2: Is there any standard protocol of VLU treatment for the patients found in the hospital or somewhere? Or the clinical medical doctors have provided the details of the VLU treatment procedures to the patients?

Response 2: We have added the following sentence to the text:

There is no standardized VLU treatment protocol on national, primary, secondary, and tertiary level, clinical medical doctors decide according to their education. Therefore, clinicians and members of their teams verbally explaine VLU treatment procedure to patients.

Point 3. Is there any difference of the VLU healing results between the different treatments (the wound was treated by the doctors/nursing staff or patient self?

Response 3: We added the following sentence to the text:

In Croatia, nurses are not allowed to prescribe treatment or wound dressing; they can follow doctor’s recommendations. Also, doctors rarely proceed wound dressing change, it is done mainly by nurses or patients.

However, our focus was on patients’ experiences and activities rather than wound healing rates.

Point 4. Are dressing types the same used for all the patients or not? How many types of the dressing were used by the patients? Did the different dressings show the different healing results?

Response 4. For clarification, we included the table in the Supplement section (Table S1).

Point 5. How about the age effect on self-treated VLU? (The younger patients did better or worse?)

Response 5. For a better understanding, we included the following text:

However, most participants thought they were old and had to be sick and were surrendered to the disease as such, especially if their family had someone with a venous ulcer. We found that older couples were more connected and took care of each other, while younger patients expressed more anger or loneliness, similar to a previous study.

Point 6. Table 1 should be re-do. Please use “.” To instead of “,” in all the numbers in Table 1, such as “68,2” should be “68.2”; “7,6” should be “7.6”; “37,5” should be “37.5”; “62,5” should be “62.5”; ect.

Response 6. Thank you for your suggestion. We followed the recommendations and changed all numbers accordingly.

Point 7. The authors should give their suggestions about self-treatment. I am just wondering if the authors recommend the VLU patients to self-treat the VLU? What is the authors’ suggestion?

Response 7. We believe that patients can participate in wound care in accordance with their abilities, but that is why they need instruction from trained staff. Our suggestion is to empower patients by educational approaches to follow the recommendations of physicians and nurses, physical activities included. Patients must be able to both obtain and fully understand such information.

Point 8. The study results should be described by more tables or graphs, rather than a text description.

Response 8. Thank you for this suggestion. We included Tables 3. and 4 in Results section.

We hope that this changes will fulfill your requirements. Thank you very much in advance.

1. Guest G, Bunce A, Johnson L. How Many Interviews Are Enough? An Experiment with Data Saturation and Variability.[J]. Field Methods, 2006, 18(18):59-82. Doi: 10.1177/1525822X05279903.

2. Coyne IT. Sampling in Qualitative Research. Purposeful and Theoretical Sampling; Merging or Clear Boundaries? J Adv Nurs. 1997;67:623–630.

3. Hareendran, A.; Bradbury, A.; Budd, J.; Geroulakos, G.; Hobbs, R.; Kenkre, J.; Symonds, T. Measuring the Impact of Venous Leg Ulcers on Quality of Life. J Wound Care 2005, 14 (2), 53–57. https://doi.org/10.12968/jowc.2005.14.2.26732.

Reviewer 2 Report

Thank you for the opportunity to revise the manuscript “Wounds home alone” - experiences of venous leg ulcer patients with self-treatment practice: A qualitative content study.

It is an interesting topic, but to give the study greater potential, it should have been given a different approach. It is too focused on the health care aspect, and only in one specific country, which limits the study and prevents it from being as novel as it could be.  I would advise, instead of focusing on why and how patients carry out their self-treatment in ulcer care, focusing on the experience of those patients, as the title would suggest.

On the other hand, the study has serious methodological deficiencies that need to be taken into account. 

It is considered a qualitative study, and the title itself is very compelling, but upon reading more in depth and learning the procedure carried out for the study, it is not actually this type of research study.

Firstly, it lacks a qualitative methodological focus that guides the analysis of the theory.  Regarding sampling, neither the procedure nor the technique used in selecting the sample are mentioned.  The goal is to gain a deeper understanding of the participants' experiences, but only short semi-structured interviews have been carried out.  This is another one of the limitations. What questions were asked to the subjects? The interview length seems very short to have obtained sufficient information.

The results describe what the patients said, but they do not look deeper into the patients' experiences. Some of the dimensions are not appropriate for a qualitative study.

The discussion needs to be more in-depth, as the author has simply given a description of the results.

Author Response

Response to Reviewer 2 Comments

Point 1: Thank you for the opportunity to revise the manuscript “Wounds home alone” - experiences of venous leg ulcer patients with self-treatment practice: A qualitative content study.

Response 1: Thank you very much for your effort and time reviewing this manuscript. We appreciate all of your comments.

We have changed the title to “Wounds home alone – why and how venous leg ulcer patients self-treat their ulcers: A qualitative content study.

Point 2: It is an interesting topic, but to give the study greater potential, it should have been given a different approach. It is too focused on the health care aspect, and only in one specific country, which limits the study and prevents it from being as novel as it could be. 

Response 2. As this is one of the first studies about venous leg ulcers self treatment, we view this almost as an introduction to the topic and have limited its scope accordingly. We look forward to expanding the scope of this research internationally. To improve the depth of the study, however, we have added some aspects about the  psychosocial dimensions of care. For example, we have add the following parts to the paper:

·         Patients quotations in subtheme”Fear from the future”

·         Discussed age effect in self-treat

·         Added table which illustrates role of informal caregiver

·         Discussed patient -home care nurse realtionship

Point 3. I would advise, instead of focusing on why and how patients carry out their self-treatment in ulcer care, focusing on the experience of those patients, as the title would suggest.

Response 3: Thank you for pointing this. We have changed the title to be better in line with the content of the manuscript.

Point 4. On the other hand, the study has serious methodological deficiencies that need to be taken into account.

It is considered a qualitative study, and the title itself is very compelling, but upon reading more in depth and learning the procedure carried out for the study, it is not actually this type of research study.

Firstly, it lacks a qualitative methodological focus that guides the analysis of the theory.

Response 4. This study is part of the doctoral research of one of the authors (Mirna Žulec), a postgraduate student at the Medical Faculty of the University of Ljubljana, Slovenia. The governmental body of the university approved the doctoral thesis together with a qualitative study, which was presented and approved in way it is proceeded in this manuscript. We have followed various published recommendations for qualitative research, and believe that our study does indeed meet the criteria of a qualitative study [1–3].

Point 5. Regarding sampling, neither the procedure or the technique used in selecting the sample are mentioned.

Response 5. Thank you for the opportunity to clarify this issue. Criterion sampling is mentioned in the sampling section.

Criterion sampling has the following characteristics [4, 5]:

• can be useful for identifying and understanding cases that are information-rich

• can provide an important qualitative component to quantitative data

• can be useful for identifying cases from a standardized questionnaire that might be useful for follow-up

For clarification and technique we added additional text to Sampling section.

Point 5.  The goal is to gain a deeper understanding of the participants' experiences, but only short semi-structured interviews have been carried out.  This is another one of the limitations. What questions were asked to the subjects? The interview length seems very short to have obtained sufficient information.

Response 6. Semi-structured questionnaire can be found in Supplement file. During interview participants were encouraged to answer about self care as interviewer asked: “Please describe...”, “What do you mean...”, “How did you felt...”, etc. Regarding interview time, it can be very individually, but after few interviews, researcher found the optimal time and way to go through interview with keeping on mind that participants are old and ill people. Longer interview could get the opposite effect and make participant not willing to participate in research.

Point 7. The results describe what the patients said, but they do not look deeper into the patients' experiences. Some of the dimensions are not appropriate for a qualitative study.

The discussion needs to be more in-depth, as the author has simply given a description of the results.

Response 7. Thank you for this suggestion, we changed the Discussion section to be more in-depth.

1.      Http://Www.Equator-Network.Org/?Post_type=eq_guidelines&eq_guidelines_study_design=qualitative-Research&eq_guidelines_clinical_specialty=0&eq_guidelines_report_section=0&s=.

2.      Consolidated criteria for reporting qualitative research (COREQ): a 32-item checklist for interviews and focus groups | International Journal for Quality in Health Care | Oxford Academic https://academic.oup.com/intqhc/article/19/6/349/1791966 (accessed Jan 28, 2019).

3.      Https://Academic.Oup.Com/Intqhc/Article/19/6/349/1791966.

4.      Palinkas, L. A.; Horwitz, S. M.; Green, C. A.; Wisdom, J. P.; Duan, N.; Hoagwood, K. Purposeful Sampling for Qualitative Data Collection and Analysis in Mixed Method Implementation Research. Adm Policy Ment Health 2015, 42 (5), 533–544. https://doi.org/10.1007/s10488-013-0528-y.

5.      Qualitative Research Guidelines Project http://www.qualres.org/ (accessed Jan 28, 2019).

Reviewer 3 Report

This is a very interesting article, in particular by identifying some of the issues that lead to the: “Wounds home alone”.

The methodological approach has some limitations. An exclusively qualitative approach is essentially used to study preceptions or experiences about a phenomenon, and this study went further than this.

A more quantitative approach to issues such as, for example:

1. How long do you have ulcer/ulcer? Is it the first time, or you had it before?

5. What is the local treatment in the last 30 days?

6. Do you apply compression therapy? What kind?

10. Where you get information about your ulcer therapy?

11. Does someone help you with self-treat? Who?

And a more qualitative approach to the issues:

2. What do you think is the cause of ulcer?

6. (…) you know what the purpose of compression therapy is?

8. Do you self-treat your wound? When and why? Please describe what you do?

9. Did you sometimes think about changing local wound therapy by yourself? Please let me know why? What did you do?

11. (…)  How he/she helps you?

12. In overall, what do you find as the biggest obstacle/problem in healing your wound and generally in wound healing?

Throughout the text, some references (relative frequencies) appear for some passive responses of a more quantitative approach, such as those described above. An example of this, is in Table S1, which reflects the answers to questions 3, 4 and 5, of the semi-structured questionnarie.

Author Response

Response to Reviewer 3 Comments

Point 1: This is a very interesting article, in particular by identifying some of the issues that lead to the: “Wounds home alone”.

Response 1: Thank you very much for your nice comment and for recognizing our research as an interesting topic. We appreciate all of your comments.

Point 2: The methodological approach has some limitations. An exclusively qualitative approach is essentially used to study preceptions or experiences about a phenomenon, and this study went further than this.

Response 2: As one of the first studies on venous leg ulcer self treatment, we felt that introducing some facts about the illness and recent wound care would be necessary and beneficial to better explain the theme of the research/paper. We agree that the study went further. This was mainly due to the reported experiences of patients, which brought up organizational problems, lack of protocols, and other issues that we then felt were necessary to address. 

Point 3. A more quantitative approach to issues such as, for example:

1.      How long do you have ulcer/ulcer? Is it the first time, or you had it before?

Response 3: These data are presented in Table 1.

Point 4. What is the local treatment in the last 30 days?

Response 4: Tahnk you for this question. This was very interesting topic and answers are presented in the Supplemental file.

Point 5. Do you apply compression therapy? What kind?

Response 5: It depended on various factors. We added Table 3 to better understand use of compression therapy on participants.

Point 6.  Where you get information about your ulcer therapy?

Response 6: Patients only received verbal instructions from healthcare providers, nurses, and doctors, we included this information into the text.

Point 7. Does someone help you with self-treat? Who?

Response 7: Informal caregivers (significant others) become interesting theme in recent researches in chronic conditions. We found that one third of participants had the help of family members. We added Table 4 for more details.

Point 8. Throughout the text, some references (relative frequencies) appear for some passive responses of a more quantitative approach, such as those described above. An example of this is in Table S1, which reflects the answers to questions 3, 4 and 5, of the semi-structured questionnaire.

Response 8. Although our study was mainly qualitative in nature, we did want readers to have an idea of the frequencies and trends of self-care of wounds among our sample population; we feel that such information provides a fuller account of our findings and thus a more robust research report. If the journal does not object to it, we would like to retain this information in the paper.

Round  2

Reviewer 1 Report

The authors have addressed most of the questions, and the revised manuscript is much better than the old version. However, there still has some mistakes should be corrected including some typing mistakes. The Table 4 should be in one page (don't separate it into two pages).

Reviewer 2 Report

The manuscript has been improved for publication.